# Detecting Phenological Development of Winter Wheat and Winter Barley Using Time Series of Sentinel-1 and Sentinel-2

**Katharina Harfenmeister** [1],*[iD], **Sibylle Itzerott** [1][iD], **Cornelia Weltzien** [2,3][iD] **and Daniel Spengler** [1][iD]

1  Helmholtz Centre Potsdam, GFZ German Research Centre for Geosciences, Telegrafenberg,
   14473 Potsdam, Germany; itzerott@gfz-potsdam.de (S.I.); daniel.spengler@gfz-potsdam.de (D.S.)
2  Technische Universität Berlin, Chair of Agromechatronics, Straße des 17. Juni 144, 10623 Berlin, Germany;
   cweltzien@atb-potsdam.de
3  Leibniz Institute for Agricultural Engineering and Bioeconomy (ATB), Max-Eyth-Allee 100,
   14469 Potsdam, Germany
*  Correspondence: katharina.harfenmeister@gfz-potsdam.de; Tel.: +49-331-288-28775

**Abstract:** Monitoring the phenological development of agricultural plants is of high importance for farmers to adapt their management strategies and estimate yields. The aim of this study is to analyze the sensitivity of remote sensing features to phenological development of winter wheat and winter barley and to test their transferability in two test sites in Northeast Germany and in two years. Local minima, local maxima and breakpoints of smoothed time series of synthetic aperture radar (SAR) data of the Sentinel-1 VH (vertical-horizontal) and VV (vertical-vertical) intensities and their ratio VH/VV; of the polarimetric features entropy, anisotropy and alpha derived from polarimetric decomposition; as well as of the vegetation index NDVI (Normalized Difference Vegetation Index) calculated using optical data of Sentinel-2 are compared with entry dates of phenological stages. The beginning of stem elongation produces a breakpoint in the time series of most parameters for wheat and barley. Furthermore, the beginning of heading could be detected by all parameters, whereas particularly a local minimum of VH and VV backscatter is observed less then 5 days before the entry date. The medium milk stage can not be detected reliably, whereas the hard dough stage of barley takes place approximately 6–8 days around a local maximum of VH backscatter in 2018. Harvest is detected for barley using the fourth breakpoint of most parameters. The study shows that backscatter and polarimetric parameters as well as the NDVI are sensitive to specific phenological developments. The transferability of the approach is demonstrated, whereas differences between test sites and years are mainly caused by meteorological differences.

**Keywords:** phenology; agriculture; Sentinel-1; Sentinel-2; time series

## 1. Introduction

The adaption of agricultural production to climatic changes to ensure global food security as well as the simultaneous conservation of the environment in times of increasing land scarcity is one of the main tasks of agriculture today [1–3]. In this context, the knowledge about prevailing phenological conditions of agricultural crops is of high importance for farmers. Monitoring the phenological development of agricultural crops enables farmers to predict crop yield and react to unfavorable conditions to a certain degree, for example, irrigation at dry conditions, adapting the harvest date in case of estimated rain or with site-specific fertilization strategies [4–6]. Additionally, researchers from varying fields are interested in current and historical data of crop phenology, exemplary as input parameters for hydrological or climate models [7,8].

The phenological development describes the life cycle of a plant which is initiated and influenced by environmental changes and is highly influenced by current weather conditions [9]. The BBCH scale (named after its developing institutes Biologische Bundesanstalt, Bundessortenamt, und CHemische Industrie) is a system to standardize phenologically

similar growth stages of multiple plant species and is used worldwide by research and administration [9].

The monitoring of phenological development as well as the detection of specific phenological stages of agricultural plants using remote sensing data is widely used [10]. Additionally, time-lapse cameras installed close to the surface (PhenoCams) are used to track phenology at very high temporal resolutions (e.g., at a daily rate) but are mostly limited to the field scale [11,12]. The great advantage of remote sensing data is their ability to cover large areas in regularly time steps. Furthermore, images from several sensors are available free of charge and have been acquired for many years. Data from optical sensors like Landsat, Sentinel-2 or MODIS and particularly derived vegetation indices like NDVI (Normalized Difference Vegetation Index) are often and successfully used to monitor agricultural plants and to detect their phenological development in the past [13–15].

However, a main disadvantage of optical remote sensing data is their dependence on cloudless conditions. In contrast, synthetic aperture radar (SAR) data is independent of cloud cover and turned out to be sensitive to crop parameters like biomass, plant height and leaf area index (LAI) as well. Strong correlations between SAR data and biophysical parameters of agricultural crops has been shown by several studies mostly for C-Band [16–21], but also for other SAR wavelengths like X-Band [22,23] or L-Band [24–26]. In addition, many studies investigated the sensitivity of time series of backscatter and polarimetric decomposition parameters to phenological changes like heading or harvest [27–29].

Furthermore, time series of SAR data like those of Sentinel-1 are suitable to monitor phenological development of agricultural crops. Schlund and Erasmi [30] and Löw et al. [31] used Sentinel-1 time series for VV and VH backscatter as well as interferometric coherence to detect entry dates of phenological stages. Additionally, Löw et al. used alpha, entropy and several elements derived from a Kennaugh Matrix, whereas Schlund and Erasmi concentrated on wheat fields of the years 2017, 2018 and 2019, Löw et al. additionally included sugar beet and canola fields to the analyses but concentrated on the year 2017. Each study was performed for one test site in Germany. Furthermore, Nasrallah et al. [32] monitored phenological development of winter wheat in Lebanon using backscatter time series of Sentinel-1 which are smoothed using a Gaussian filter and compared it to NDVI time series. Seeding and harvest dates of several agricultural crops could also be detected by using Sentinel-1 coherence of interferometric SAR data [33]. Furthermore, it was found that Sentinel-1 and Sentinel-2 data (VH/VV and NDVI) are sensitive to crop growth particularly for major European winter crops [34]. The combination of Sentinel-1 and Sentinel-2 data was also used by Mercier et al. [35] to predict phenological stages of wheat and rapeseed using a classification approach.

Most previous studies focused on single years [31,34] and/or on single test sites [30,32]. Therefore, this study focuses on the transferability of the detection of phenological entry dates by comparing two test sites and two years. Furthermore, this study includes winter barley in the analyses, a crop type that was not considered in previous studies. Additionally, time series of NDVI derived from Sentinel-2 are compared to the results of Sentinel-1 time series to evaluate the strengths and deficiencies of optical and SAR sensors. SAR parameters are sensitive to structural changes of the plants as well as to moisture changes and to the variable contribution of vegetation and soil to the signal which leads to changing scattering mechanisms [16,17]. The NDVI is mainly sensitive to photosynthetic activity and consequently also detects changes in soil vegetation contributions [15].

The aim of this study is to identify phenological entry dates of winter wheat and winter barley using smoothed time series of the Sentinel-1 radar backscatter parameters VH, VV and their ratio VH/VV as well as of the three parameters entropy, anisotropy and alpha derived from polarimetric decomposition. In addition, the vegetation index NDVI derived from Sentinel-2 data is included in this study to incorporate vegetation vitality information. Local minima, local maxima and breakpoints of the time series are compared with phenological entry dates reported by the German Weather Service (Deutscher Wetterdienst (DWD)) for two test sites in Northeast Germany. As a result,

differences between reported phenological entry dates and metrics of the time series are presented as number of days and discussed in detail.

## 2. Study Sites

The analyzed wheat and barley fields are located in two study sites in Northeast Germany (Figure 1). Both regions are characterized by glacio-fluvial landforms like extensive, flat sand regions, hills and sinks.

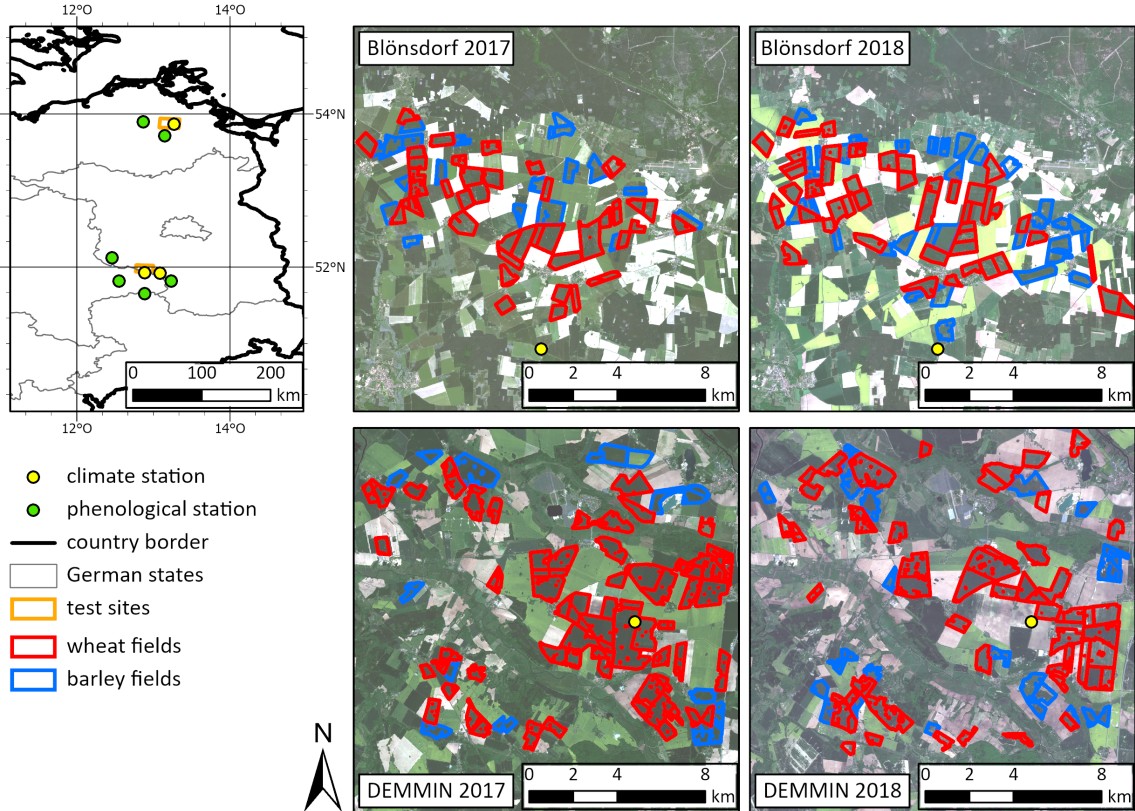

**Figure 1.** Location of the test sites DEMMIN and Blönsdorf in Germany and location of observed wheat and barley fields. Background: Sentinel-2 images of 29 May 2017 and 7 May 2018 (WGS 1984, UTM Zone 32N).

The test site *DEMMIN* (Durable Environmental Multidisciplinary Monitoring Information Network) is located in the federal state Mecklenburg-West Pomerania and is part of several monitoring projects like JECAM (Joint Experiment of Crop Assessment and Monitoring) [36] and TERENO (Terrestrial Environmental Observatories) [37,38]. The test site *Blönsdorf* is located approximately 200 km south of DEMMIN in the federal state Brandenburg in a range of hills called "Fläming Heath".

The climate in both test sites belongs to the transition zone between continental and maritime climate with mean annual temperatures of 8.8 °C (DEMMIN) and 9.3 °C (Blönsdorf). The total annual precipitation is about 600 mm in DEMMIN and 540 mm in Blönsdorf [39]. The two analyzed years 2017 and 2018 are meteorologically very different with remarkably lower precipitation sums and higher temperatures in 2018 (Figure 2).

Both test sites are extensively agriculturally used with mainly conventionally managed fields. Fertilizers and pesticides are commonly used to increase yields, while precision agriculture is only used in individual cases.

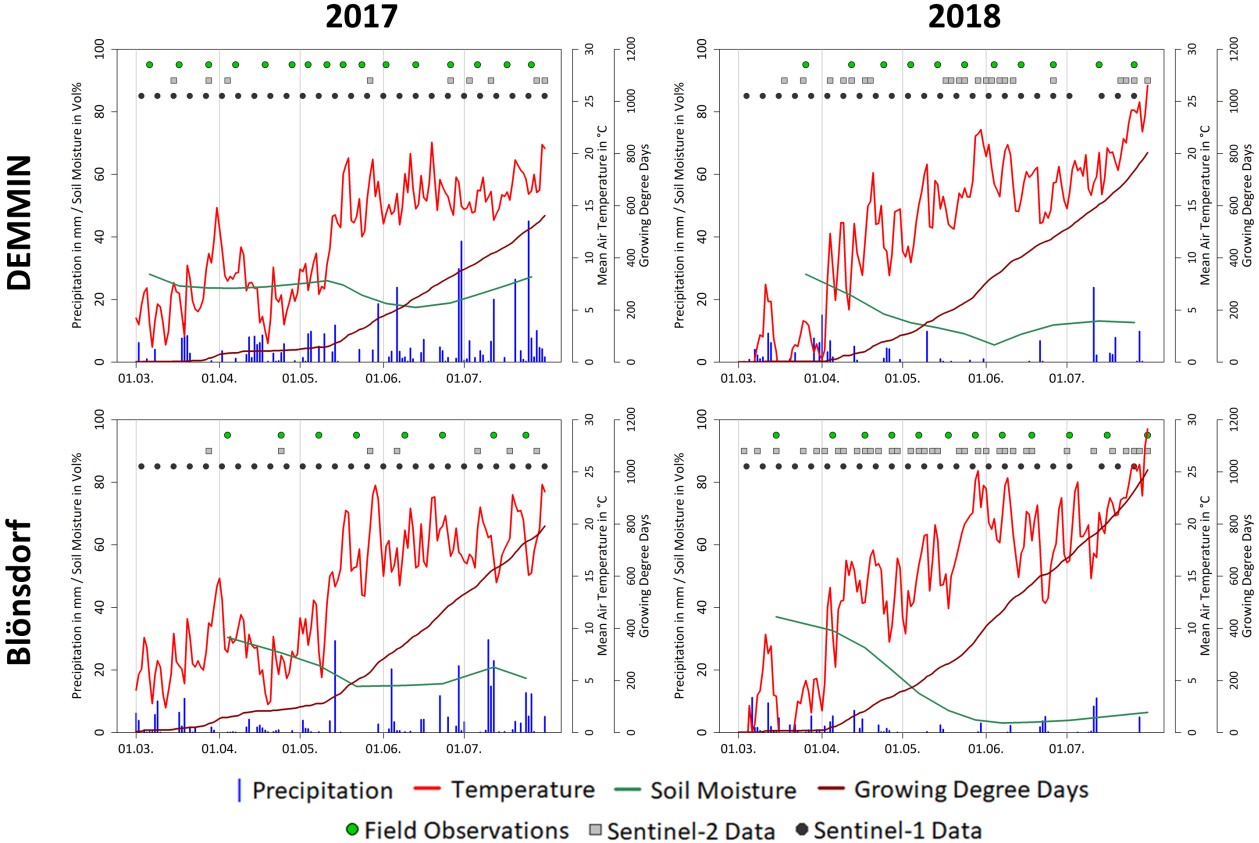

**Figure 2.** Meteorological differences between test sites and years and data availability of Sentinel-1 and Sentinel-2 as well as dates of field observations.

## 3. Materials and Methods

### 3.1. Data

#### 3.1.1. Sentinel-1 Data

The two identical Sentinel-1 satellites operated by the European Copernicus Program of the ESA (European Space Agency) provide synthetic aperture radar (SAR) data with a high revisit frequency of six days with equal acquisition conditions (equal pass, orbit and incidence angle). The C-band instrument (5.405 GHz) offers dual polarized SAR data in VH (vertical-horizontal) and VV (vertical-vertical) polarization. It has a 250 km swath, a range spacing of 2.33 m and an azimuth spacing of 13.89 m.

In this study, Sentinel-1 data from the ascending orbit 146 with a mean local incidence angle of ~38° (DEMMIN) and ~34° (Blönsdorf) is used because of a limited comparability of Sentinel-1 images from different orbits and pass directions [16]. In total, 100 Sentinel-1 images from both years 2017 and 2018, each from the beginning of March until the end of July, and from both test sites DEMMIN and Blönsdorf are used in this study to create time series of backscatter and polarimetric parameters (Table 1 and Figure 2).

**Table 1.** Available Sentinel-1 and Sentinel-2 data and analyzed fields per test site and per year. Values in parentheses indicate Sentinel-2 data suitable for the majority of fields.

| Test Site | Year | Sentinel-1 | Sentinel-2 | Wheat Fields | Barley Fields |
|---|---|---|---|---|---|
| DEMMIN | 2017 | 26 | 9 (8) | 59 | 15 |
| | 2018 | 24 | 22 (18) | 51 | 22 |
| Blönsdorf | 2017 | 26 | 7 (6) | 39 | 18 |
| | 2018 | 24 | 35 (29) | 46 | 33 |

### 3.1.2. Sentinel-2 Data

Sentinel-2A and Sentinel-2B are two identical satellites operated by ESA. They provide multispectral images with a high spatial resolution up to 10 m and a high temporal resolution of two to three days. In this study, 73 Sentinel-2 images are used (Figure 2). The majority of them were acquired in 2018 (57 images). Because of partial cloud coverage, some images are not covering all analyzed fields (Table 1). The red and the near infrared band of the Sentinel-2 sensors are used to calculate time series of the vegetation index *NDVI* (Equation (1)).

$$NDVI = \frac{NIR - Red}{NIR + Red} \tag{1}$$

### 3.1.3. Field Data

In total, time series of 195 winter wheat and 88 winter barley fields are analyzed in both years and both test sites (Table 1 and Figure 1). Thirteen of these fields were regularly visited during the vegetation periods of 2017 and 2018, and these serve as reference fields to validate phenological data reported by the German Weather Service (DWD). Field boundaries and crop types for each year were extracted from the IACS (integrated administration and control system) data bases of Mecklenburg-West Pomerania and Brandenburg [40]. Field sizes range between 20 ha and 234 ha with an average field size of 50 ha.

### 3.1.4. Meteorological and Phenological Data

Phenological entry dates are reported immediately or annually by volunteer observers at numerous phenological observation stations in Germany operated by the German Weather Service (DWD) [41]. In this study, observations of the phenological stations closest to the fields of the test sites are used. The plausibility of the reported entry dates was verified using own phenological observations. In total, observations from four phenological stations for each test site were used for the analyses. These were for DEMMIN the stations Tützpatz (ID 12508) and Dargun (ID 12615) with data from both annual and immediate reporters. The stations of the immediate reporter of Dommitzsch (ID: 13005) as well as the annual reporters of the stations Wiesenburg (ID: 5546), Schönewalde (ID: 12306) and Selbitz (ID: 13102) were used for the Blönsdorf test site (Figure 1).

The phenological observations of DWD report the entry dates of five different phenological development stages for winter wheat and winter barley, which are directly related to the BBCH system [9] (Table 2 and Figure 3). One main difference between the descriptions of the phenological stations is the DWD specification of how many plants of a field should already show a specific development stage. In contrast, BBCH stages refer to individual plants and do not make any statements about the percentage of a field that should undergo a specific phenological stage.

Meteorological data are provided by DWD for the Blönsdorf test site by the stations Naundorf bei Seyda (ID: 03445, precipitation) and Langenlipsdorf (ID: 02856, air temperature) [39]. For the DEMMIN test site, the climate station Heydenhof of the TERENO meteorological network provided precipitation and air temperature data [42]. The climate stations provide daily precipitation sums as well as daily air temperatures and were used to investigate phenological differences between the two observed years 2017 and 2018 (Figure 2).

**Table 2.** Phenological stages reported by DWD and their related BBCH stage.

| DWD | BBCH | Name | Description |
|---|---|---|---|
| 15 | 31 | Beginning of Stem Elongation | About half of the plants grow clearly in length and the first stem node above the ground is perceptible. |
| 18 | 51 | Beginning of Heading | At about half of the stems, the first spikelets are visible and emerge laterally from the sheats. |
| 19 | 75 | Medium Milk | The grain content is milky. The first grains reached their final size and are still green. |
| 21 | 87 | Hard Dough | First grains in about half of the ears have changed their color from green to yellow and can be easily removed from from the panicle. The grain content is solid. |
| 24 | 99 | Harvest | The field is harvested. |

### 3.2. Methods

#### 3.2.1. Data Processing

Sentinel-1 images were downloaded as SLC data (single look complex) in IW mode (interferometric wide swath mode). Before calibration, all images were split and the appropriate orbit file was applied using the Sentinel Application Platform (SNAP) [43]. To extract the backscatter coefficients of VV and VH, the data were calibrated to $\sigma^0$ and debursted. Afterwards, all images were multi-looked and smoothed using the "Refined Lee" speckle filter with a window size of $7 \times 7$.

To perform the polarimetric decomposition, Sentinel-1 images were calibrated to a complex output and debursted. The polarimetric speckle filter "Refined Lee" with a window size of $5 \times 5$ was applied to enhance image quality [44]. The H-*alpha* dual polarimetric decomposition algorithm with a window size of $5 \times 5$ was used to extract entropy, anisotropy and alpha [45,46]. Entropy is a measure of the randomness of the scattering process, anisotropy indicates the presence of a second scattering mechanism and alpha describes the dominant scattering mechanism [47]. Resulting alpha values were subtracted from $90°$ to get the appropriate value range. It has to be considered that polarimetric decomposition parameters derived from dual-polarimetric data differ from those derived from quad-polarimetric data [17,46].

Range-Doppler Terrain Correction using the digital elevation model of SRTM (Shuttle Radar Topography Mission) and a coregistration using uniformly spaced ground control points (GCP) was applied to all images as a last processing step. The spatial resolution of the processed images is 10 m $\times$ 10 m for both backscatter and polarimetric decomposition parameters. Backscatter values were converted to the Decibel (dB) unit to get a logarithmic scale, and the ratio of VH and VV backscatter was calculated afterwards by subtracting VH from VV backscatter.

Sentinel-2 data were atmospherically corrected using the SICOR algorithm [48] and georeferenced using the AROSICS algorithm [49]. The NDVI was calculated according to Equation (1) for all cloud-free pixels. Clouds were identified using the cloud mask provided by the SICOR algorithm. Additionally, extreme NDVI values, namely, values lower or higher than the extremes of the lower and upper whiskers of the boxplot diagram of the field, are removed before the calculation of the mean value per field.

The resulting backscatter parameters VH, VV and VH/VV as well as the polarimetric decomposition parameters entropy, anisotropy and alpha and the NDVI values were averaged for every agricultural field, whereas the outer 20 m of each field were excluded to avoid influences by the surrounding area.

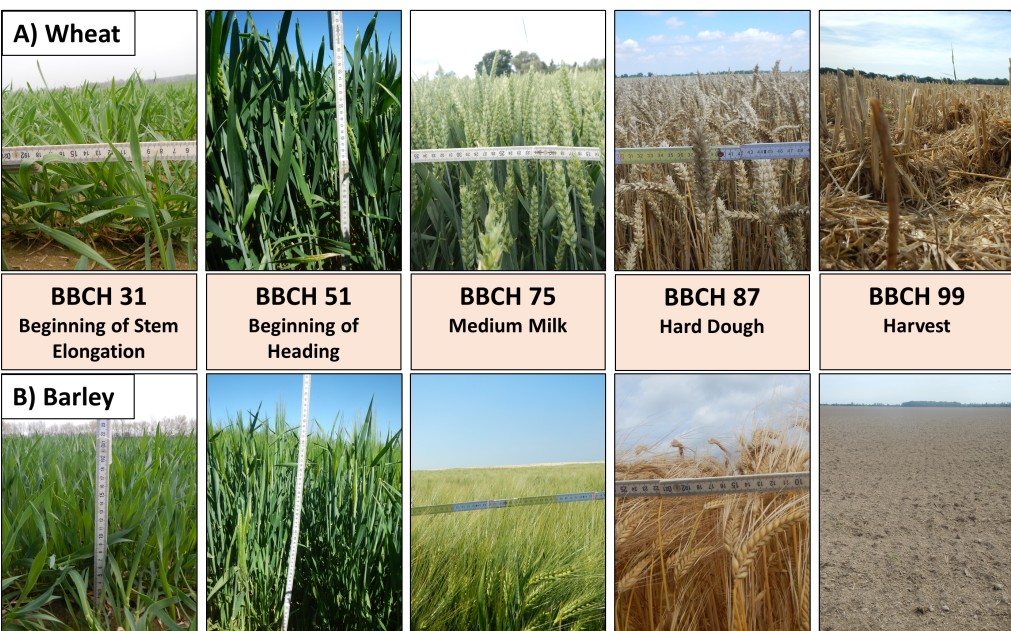

**Figure 3.** Photos of wheat (**A**) and barley (**B**) during analyzed phenological stages. Photos of the harvest stage (BBCH 99) show two exemplary fields after harvest, there are no differences between wheat and barley fields regarding field conditions after harvest.

### 3.2.2. Time Series Analysis

The time series of the extracted backscatter and polarimetric decomposition parameters as well as of the NDVI values were smoothed using the LOESS algorithm (locally estimated scatterplot smoothing) [50]. This method fits a polynomial regression for each value and its neighborhood, whereas the size of the neighborhood defines the smoothness of the fitting and is defined by the span value. The higher the span value, the more filtering of the data takes place. In this study, a one-degree polynomial regression was used. To preserve the general course of the time series, several span values were tested and visually evaluated. Finally, the span value was set to 0.3 for backscatter and the polarimetric decomposition parameters and to 0.4 for NDVI values. The higher span value of NDVI results from irregularly distributed values over time with partly large data gaps. In case only very few NDVI values were available, a smoothing was not possible and the original values were used instead. This concerns both test sites in the year 2017, when only very few Sentinel-2 images were available (Figure 2).

After smoothing the time series, local minima, local maxima and breakpoints of the fitted curve were calculated. Breakpoints were defined using the *breakpoints* function of the R-package *strucchange* [51]. The algorithm is based on Bai and Perron (2003) [52] and Zeileis et al. (2003) [53]. The main idea is to search for changing regression coefficients between segments of at least three observations by minimizing a triangular matrix of residual sum of squares (RSS) for each segment.

The dates of the detected local extrema and breakpoints were finally compared to the phenological entry dates by calculating their mean absolute differences in days (Figure 4).

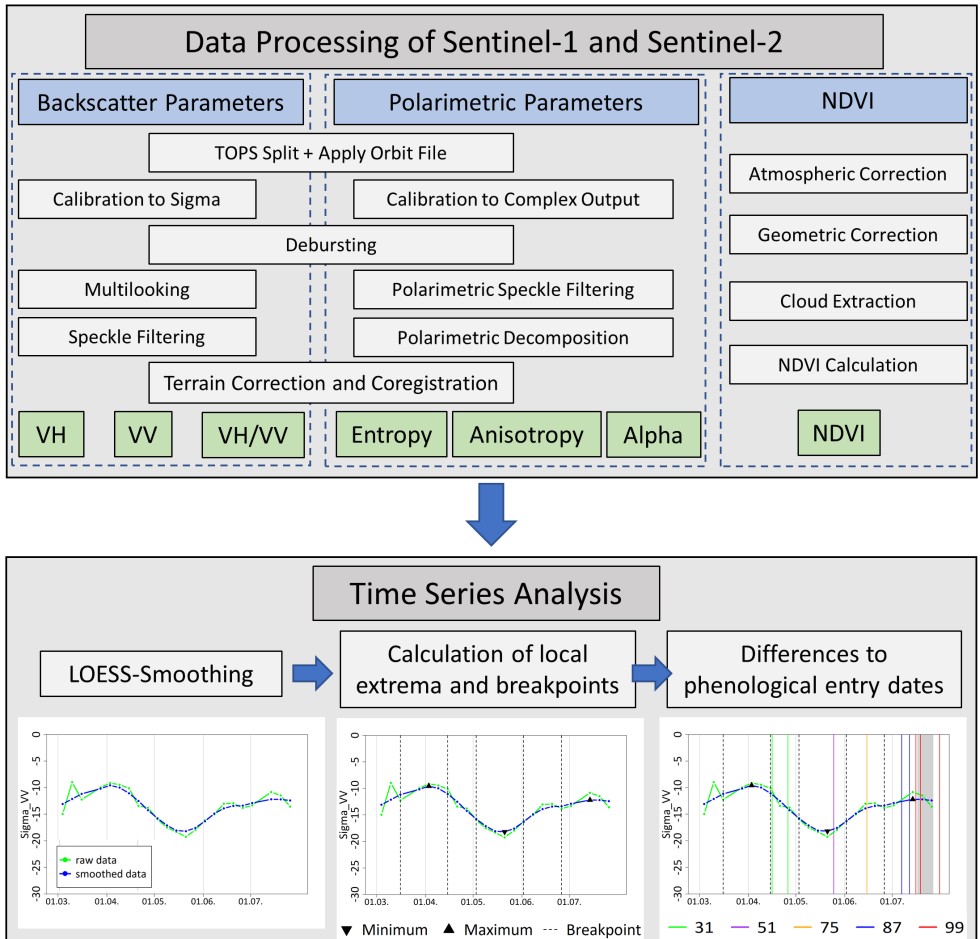

**Figure 4.** Flowchart of the data processing and time series analysis.

## 4. Results

In the following sections, results of the study are presented for each BBCH stage individually. Figures 5 and 6 show the temporal behavior of all analyzed wheat and barley fields for selected parameters, test sites and years. Boxplot diagrams within the single chapters represent parameters and metrics that best reflect the corresponding BBCH stage by having the lowest mean and median differences in days to BBCH entry dates. The lower the median value of a boxplot, the smaller is the difference in days between reported entry date and a feature of the time series of a specific parameter. The smaller the box (interquartile range), the more consistent are the results of all fields and a better transferability is assumed. At the end of Section 4, Table 3 summarizes the results of all BBCH stages and crop types.

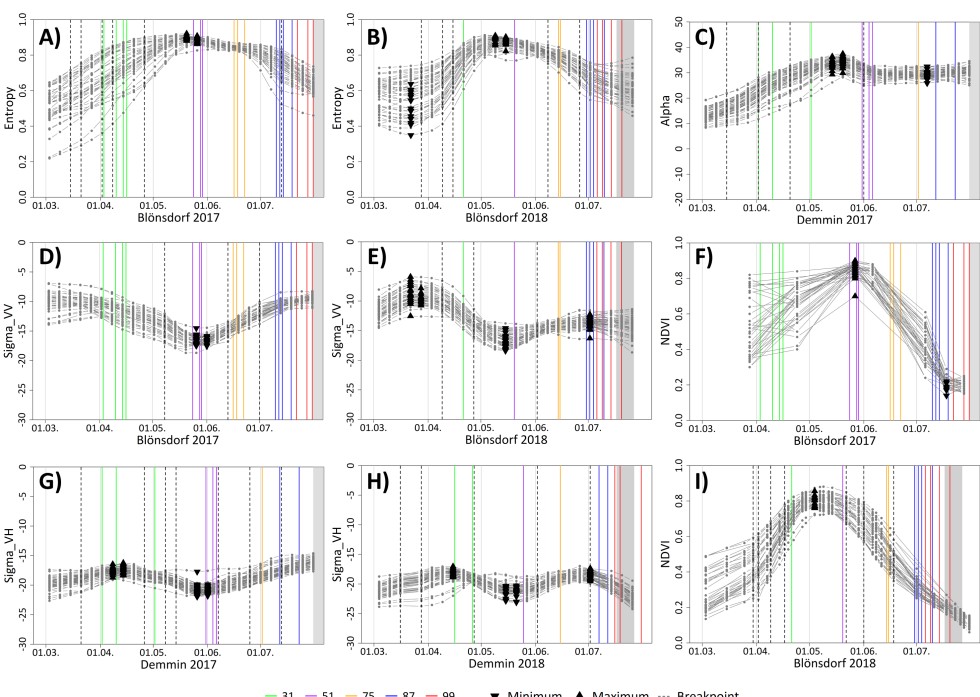

**Figure 5.** (**A–I**) Temporal behavior of selected SAR parameters and NDVI of wheat in selected test sites and years. Each gray line represents a single field. Vertical colored lines represent entry dates of BBCH stages reported by DWD phenology stations. Gray boxes mark the harvest period of own observed fields. Dashed vertical lines and triangles indicate breakpoints, local maxima and local minima which are detected by at least one third of all fields.

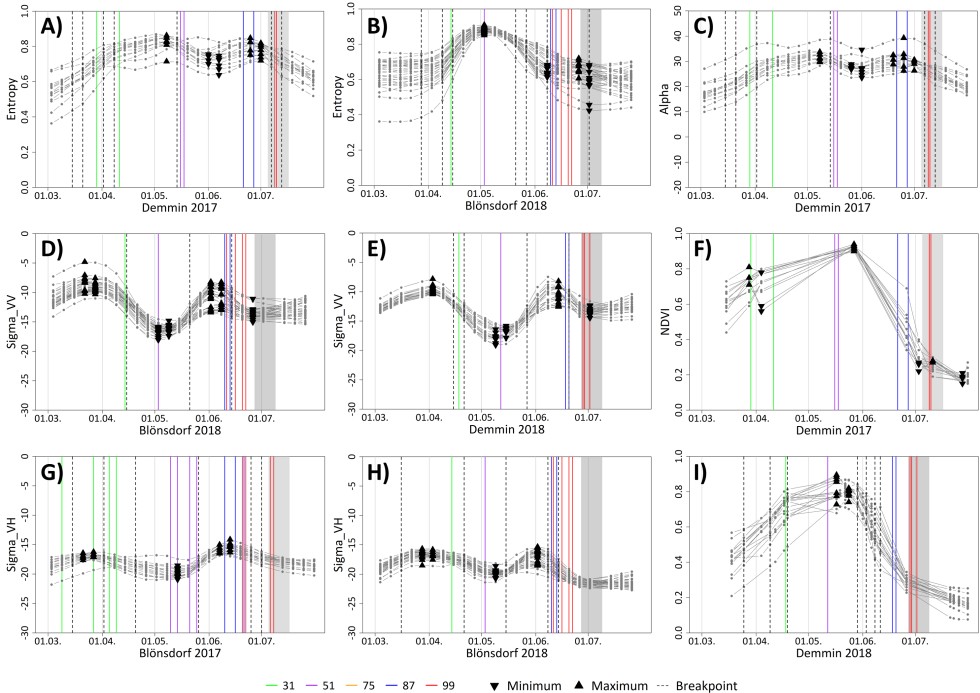

**Figure 6.** (**A–I**) Temporal behavior of selected SAR parameters and NDVI of barley in selected test sites and years. Each gray line represents a single field. Vertical colored lines represent entry dates of BBCH stages reported by DWD phenology stations. Gray boxes mark the harvest period of own observed fields. Dashed vertical lines and triangles indicate breakpoints, local maxima and local minima which are detected by at least one-third of all fields.

**Table 3.** Summarized results of all BBCH stages and crop types. The last two columns indicate mean and median differences in days between reported entry date and time series feature of the listed parameters.

| BBCH | Crop Type | Parameter | Time Series Feature | Mean | Median |
|---|---|---|---|---|---|
| 31 | Wheat | Alpha, Entropy, Anisotropy, VH/VV | 2. Breakpoint | 8–15 | 5–6 |
| | | NDVI (2018) | 2. Breakpoint | 10–14 | 4–9 |
| | | VH | Maximum | 8–15 | 6–15 |
| | Barley | VV (2017) | 1. Breakpoint | 11 | 8–10 |
| | | VV (2018) | 1. Breakpoint | 3–5 | 1–3 |
| | | Alpha, Entropy, Anisotropy, VH/VV, NDVI (2018) | 1. Breakpoint | 8 | 5 |
| | | VH (2018) | Maximum | 9 | 9 |
| 51 | Wheat | Alpha, Entropy, VH/VV | Maximum | 7–9 | 4–7 |
| | | Anisotropy | Minimum | 7–9 | 4–7 |
| | | VH, VV | Minimum | 4–7 | 2–5 |
| | | NDVI | Maximum | 2–8 | 2–7 |
| | Barley | Alpha, Entropy, VH/VV | Maximum | 9–10 | 9 |
| | | Anisotropy | Minimum | 9–10 | 9 |
| | | VH, VV | Minimum | 3–6 | 3–6 |
| | | NDVI | Maximum | 10 | 10 |
| 75 | Wheat | Alpha, Entropy, VH/VV (2017) | Minimum | 10 | 5 |
| | | Anisotropy (2017) | Maximum | 10 | 5 |
| | Barley | no data | - | - | - |
| 87 | Wheat | Alpha, Entropy, Anisotropy, VH/VV (2018) | 4. Breakpoint | 7–12 | 7–8 |
| | | VV (2018) | Maximum | 10–19 | 3–5 |
| | Barley | Alpha, Entropy, Anisotropy, VH/VV (2017) | Maximum | 7–13 | 5–7 |
| | | Alpha, Entropy, Anisotropy, VH/VV (2018) | 4. Breakpoint | 7–12 | 3–13 |
| | | VH, VV | Maximum | 6–8 | 5–9 |
| | | VH (2018) | 4. Breakpoint | 3–5 | 1–3.5 |
| | | VV (2018) | 3. Breakpoint | 6–8 | 1–3.5 |
| 99 | Wheat | not detected | - | - | - |
| | Barley | Alpha, Entropy, Anisotropy, VH/VV, VH | 4. Breakpoint | 3–12 | 2–9 |
| | | VV | Minimum | 9–14 | 2–10 |

### 4.1. BBCH 31—Beginning of Stem Elongation

The beginning of stem elongation (BBCH 31) is characterized by the erection of the main stem as well as of the tillers, whereas the top of inflorescence is located at least one centimeter above the tillering node (Figure 3). DWD phenology stations report the entry date of the beginning of stem elongation as soon as about half of the plants of the observed field grow clearly in length and their first stem node above the ground is perceptible (Table 2).

Stem elongation of wheat usually takes place at the beginning of April, in 2018 only at mid-April or at the end of April because of a delayed development caused by lower temperatures. The stem elongation of barley happens earlier than that of wheat and can already take place at the end of March (2017). In 2018, stem elongation of barley started in mid-April.

The plant appearance does not change remarkably compared to previous (end of tillering) and subsequent phases (development of further nodes during ongoing stem elongation) and the change is rather fluent. The stem elongation leads to an increase in biomass, LAI and plant height, and the radar signal is increasingly influenced by vegetation at the expense of the soil. Consequently, time series of SAR parameters as well as of NDVI do not reveal remarkable changes of the temporal behavior like local extrema around this time in April. However, multiple parameters show breakpoints during the time of the beginning of the stem elongation which confirms the growing of the plants.

Breakpoints around the entry dates of BBCH 31 are detected by NDVI and by all SAR parameters except for VV and VH backscatter (Figure 5). For wheat, the mean difference between the observed entry date of BBCH 31 and the second detected breakpoints of alpha, entropy, anisotropy and VH/VV is approximately 8–15 days. However, some extreme values influence the average difference remarkably and the median values of 5–6 days (except for Demmin 2017 with a median around 12) are much lower (Figure 7A). Breakpoints for NDVI are only calculated for 2018 because of an insufficient number of images in 2017. The mean absolute differences between the entry date of BBCH 31 and the second breakpoint of NDVI are approximately 10 days for Blönsdorf (median 4 days) and 14 days for DEMMIN (median 9 days). VV and VH backscatter show breakpoints for single fields, but no general trend is observable. Instead, VH backscatter has a local maximum around the entry dates of BBCH 31, which is particularly close in 2018 in both test sites with median values of 6 days (Figure 7B).

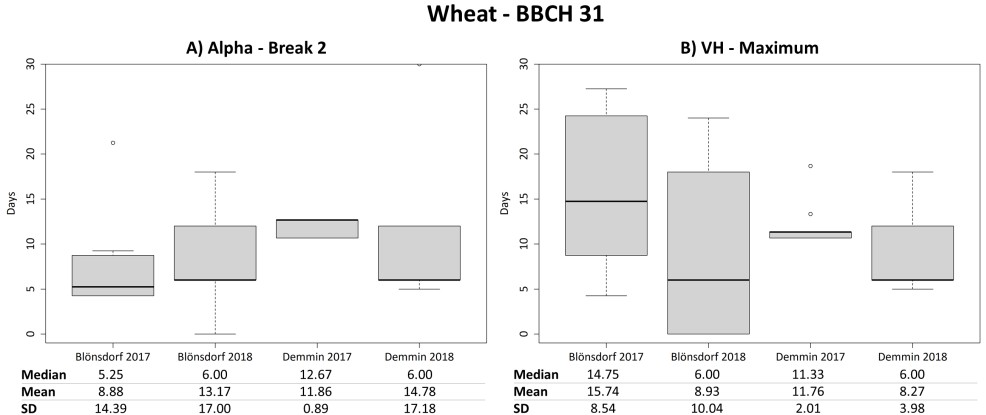

**Figure 7.** Boxplots of temporal differences in days between the second breakpoint of alpha and the entry date of the beginning of stem elongation (BBCH 31) of wheat (**A**) as well as boxplots of temporal differences in days between the local maximum of VH backscatter and the entry date of the beginning of stem elongation (BBCH 31) of wheat (**B**).

For barley, the first breakpoint instead of the second one shows lowest mean differences compared to the entry dates of BBCH 31 (Figure 6), mainly for fields of 2018. In this year, the mean difference between the observed entry dates and the first detected breakpoint is around 8 days for all SAR parameters except for VH. Median values are even lower with approximately 5 days for Blönsdorf. Particularly for VV backscatter, the first detected breakpoint and the entry date of BBCH 31 result in low mean absolute differences of approximately 4 days (median: 1 day (Blönsdorf) and 3 days (DEMMIN)) (Figure 8A). VH backscatter also shows a local maximum around the entry dates of BBCH 31 in 2018. The mean and median difference is approximately 9 days. For NDVI, the mean difference between the entry date of BBCH 31 and the first detected breakpoint of 2018 is approximately 7.5 days in Blönsdorf (median 5 days). Figure 9 displays the spatial representation of the mean difference in days between the entry date of BBCH 31 and the first breakpoint of VV backscatter for barley. In DEMMIN 2017, the second detected breakpoint instead of the first one shows the lowest mean difference of approximately 13 days (median 3.5 days).

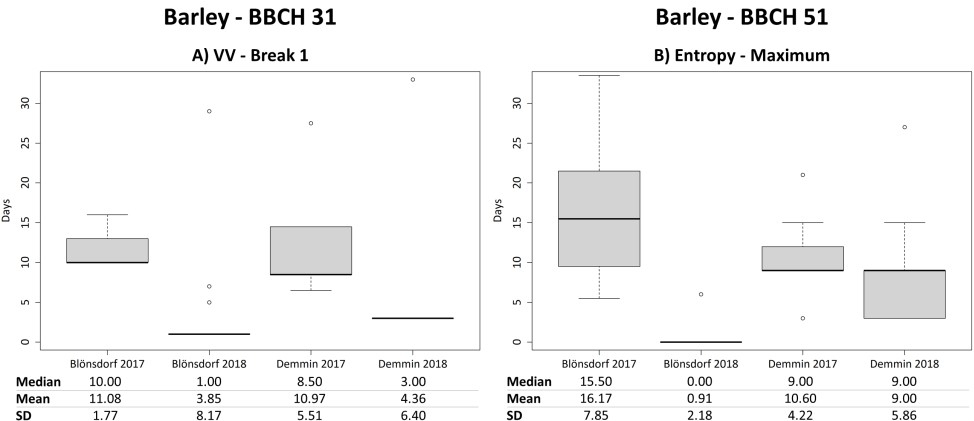

**Figure 8.** Boxplots of temporal differences in days between the first breakpoint of VV backscatter and the entry date of the beginning of stem elongation (BBCH 31) of barley (**A**) as well as boxplots of temporal differences in days between the local maximum of entropy and the entry date of the beginning of heading (BBCH 51) of barley (**B**).

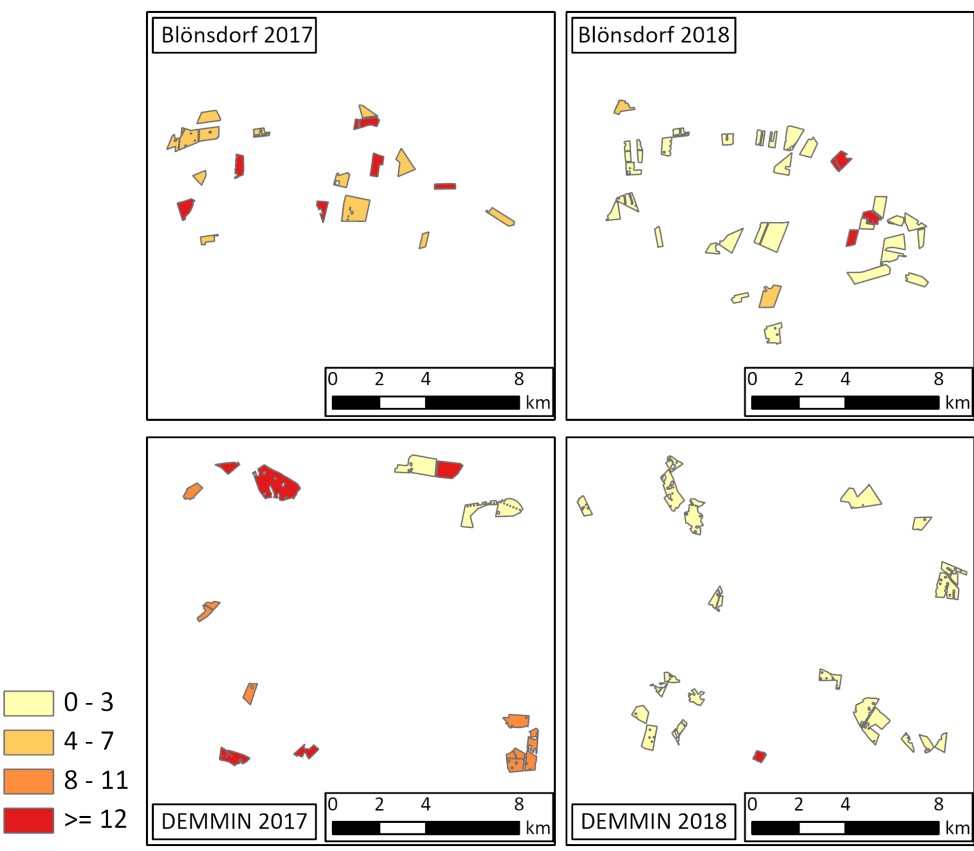

**Figure 9.** Temporal differences in days between reported entry date of BBCH 31 (beginning of stem elongation) and first breakpoint of VV backscatter of barley fields in both study areas and both years.

### *4.2. BBCH 51—Beginning of Heading*

The beginning of heading (BBCH 51) marks the time when the first spikelets are visible and emerge laterally from the sheaths (Figure 3). DWD phenology stations report the entry date of BBCH 51 as soon as about half of the stems of the observed field show the before mentioned behavior (Table 2).

The beginning of heading of wheat usually takes place at the end of May or at the beginning of June, depending on meteorological conditions of a specific year. The heading of barley starts a few weeks earlier than the heading of wheat and can already take place at the beginning of May, for example in Blönsdorf 2018.

The beginning of heading with emerging heads represents a rather strong change of the plant appearance. Biomass, LAI and plant height often reach their maximum values and the appearance of a whole field is much more divers, which particularly influences the SAR signal [16]. The analyzed SAR parameters as well as the NDVI also reflect these strong changes. All parameters show local or even global maxima or minima, which are often directly related to the heading stage.

For wheat, entropy, anisotropy, alpha and VH/VV have very similar mean absolute differences and median values between their maximum (entropy, alpha, VH/VV) or minimum (anisotropy) compared to the reported entry dates (Figure 5). Lowest mean differences of approximately 7–9 days are visible for both years and test sites except for Demmin 2017 (median values: 7 days for Blönsdorf 2017 and 4 days for Demmin 2018). The minimum values of VH and VV backscatter show even lower mean differences of around 4–7 days (median values between 2 and 5 days) for the same years and test sites (Figure 10A). Figure 11 depicts the spatial representation of the mean difference in days between the entry date of BBCH 51 and the minimum of VV backscatter for wheat.

Furthermore, the NDVI shows a global maximum around the time of the beginning of heading. The mean difference between maximum and reported entry date is particularly low for Blönsdorf 2017 and Demmin 2018 with approximately 4 days (Figure 10B).

Barley fields indicate a similar behavior as wheat. However, the polarimetric parameters entropy, anisotropy and alpha as well as VH/VV have in general higher mean and median differences of approximately 9–10 days (Blönsdorf 2017 around 16 days) between reported entry date of BBCH 51 and local maximum or minimum. Suprisingly, most Blönsdorf 2018 fields show a perfect match between local extrema and reported entry date (Figures 6B,D and 8B). The minimum values of VH and VV backscatter of barley reveal a very uniform behavior of all fields. The mean and median difference is very similar or even equal with around 3–6 days. Furthermore, NDVI maximum values are very consistent between fields, but the mean and median differences are slightly higher with around 10 days. Only Blönsdorf 2018 fields show a lower difference of approximately 4 days. Note that the local extrema of all SAR parameters are detected a few days before the entry date of BBCH 51, whereas the NDVI maximum is located a few days afterwards.

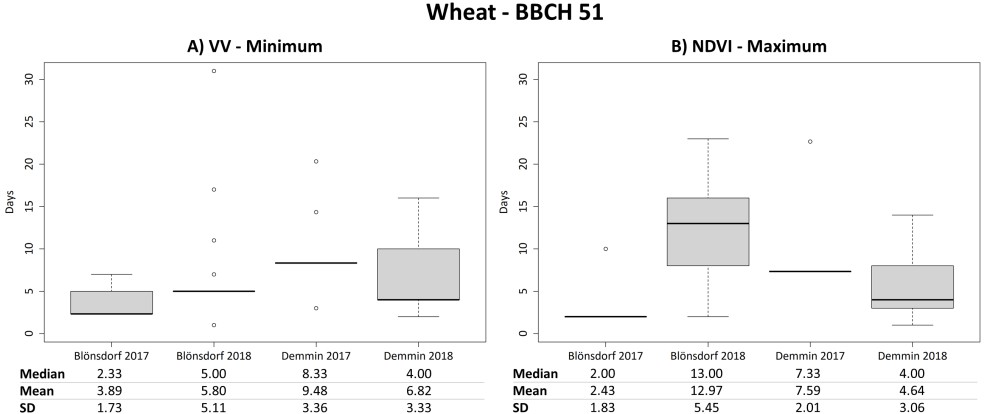

**Figure 10.** Boxplots of temporal differences in days between the local minimum of VV backscatter and the entry date of the beginning of heading (BBCH 51) of wheat (**A**) as well as boxplots of temporal differences in days between the local maximum of NDVI and the entry date of the beginning of heading (BBCH 51) of wheat (**B**).

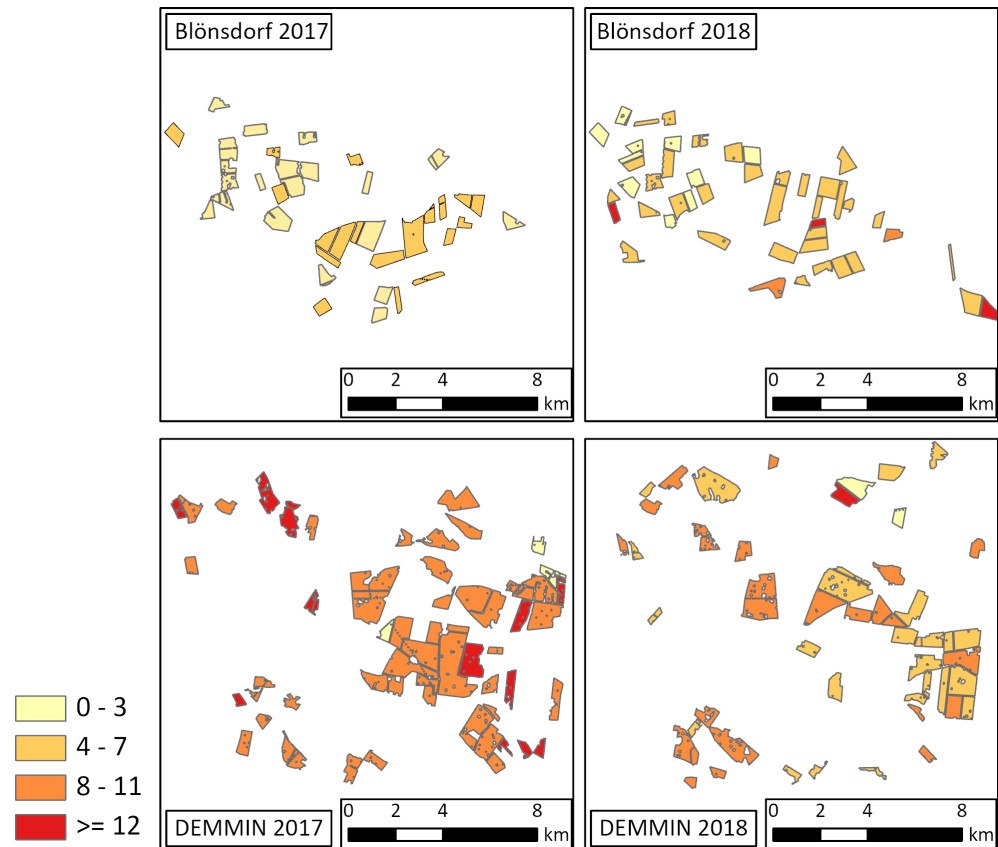

**Figure 11.** Temporal differences in days between reported entry date of BBCH 51 (beginning of heading) and local minimum of VV backscatter of wheat fields in both study areas and both years.

### 4.3. BBCH 75—Medium Milk

During fruit development, BBCH 75 indicates the stage of medium milk ripening. The grain content is still milky and first grains reached their final size while they are still green (Figure 3). The entry date of medium milk is strongly dependent on meteorological conditions of a specific year. In 2017, it started end of June (Blönsdorf) or at the beginning of July (Demmin), while in 2018 it already started in mid-June. DWD phenology stations report BBCH 75 only for wheat, therefore no data for barley is available.

The plant appearance during fruit development does not change remarkably and is rather characterized by constant or slightly decreasing biomass, LAI and plant height. Furthermore, the water content of the whole plant as well as of the grains starts to decrease as well, but is not as prominent as in the later ripening or senescence stages yet.

Consequently, there are no explicit changes of the temporal behavior of the SAR parameters and the NDVI during this time of the vegetation period (Figure 5). However, some parameters detect breakpoints or even local extrema near medium milk for single test sites or years for wheat. The polarimetric parameters entropy, anisotropy and alpha as well as VH/VV show a local minimum (entropy, alpha, VH/VV), respectively, a local maximum (anisotropy) approximately 10 days (median: 5 days) after the entry date of BBCH 75 in 2017 (Figure 12A). However, these local extrema are not that prominent because their neighboring values differ only marginally (Figure 5C). For some parameters, years and test sites, the fourth breakpoint is also detected around 10 days before the beginning of medium milk, but no consistent behavior is identified.

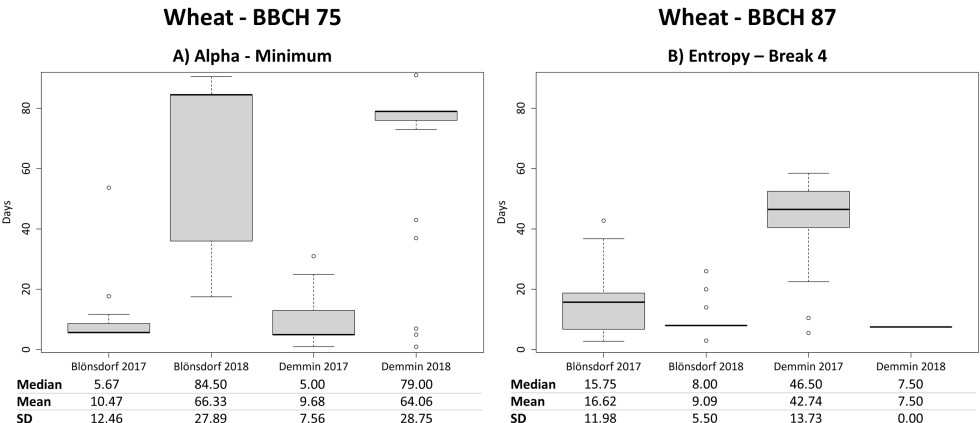

**Figure 12.** Boxplots of temporal differences in days between the local minimum of alpha and the entry date of medium milk (BBCH 75) of wheat (**A**) as well as boxplots of temporal differences in days between the fourth breakpoint of entropy and the entry date of hard dough (BBCH 87) of wheat (**B**).

*4.4. BBCH 87—Hard Dough*

BBCH stage 87 indicates the hard dough stage at the end of the ripening. DWD phenology stations report the entry date of BBCH 87 as soon as first grains in about half of the ears changed their color from green to yellow and can be easily removed from the panicle (Figure 3). The grain content is solid. For wheat, the hard dough stage starts in mid-July (2017) or already at the beginning of July (2018) depending on the meteorological conditions in the respective year. For barley, BBCH 87 already starts a few weeks earlier between mid-June and end of June.

The whole ripening stage is marked by a remarkably decreasing vegetation water content and the proceeding drying of the whole plant as well as of the grains. Their color has turned from green to yellow. The LAI decreases and the soil presumably has an increasing influence on the radar signal or the reflection again, whereas the plant volume is still higher than at the beginning of the season.

The SAR parameters usually have a breakpoint during the time of the hard dough stage for wheat in 2018 (Figure 5). In this year, the fourth breakpoint of the polarimetric parameters entropy, anisotropy and alpha as well as of VH backscatter is on average located around 7–12 days (median: 7–8 days) before the reported entry dates of BBCH 87 (Figure 12B). In 2017, the fourth breakpoint also indicates hard dough in Blönsdorf, but because not all fields show this behavior, mean and median differences in days are still high. The crop development stagnated in DEMMIN in July 2017 because of multiple heavy rain events and relatively low temperatures, therefore ripening and particularly hard dough is not detectable. VV backscatter has a local maximum before the hard dough stage in 2018 with a median difference of 3–5 days. However, this local maximum only differs marginally from its neighboring values. Furthermore, NDVI detects a local minimum at approximately 4 days after BBCH 87, but only for the Blönsdorf fields of 2017.

Differences between the two years 2017 and 2018 for BBCH 87 are also observable for barley. While polarimetric parameters as well as VH/VV show a breakpoint around 7–12 days before the hard dough stage in 2018 (median values: 3.5 days in Blönsdorf and 13 days in DEMMIN), the same parameters have a local maximum (alpha, entropy, VH/VV) or local minimum (anisotropy) with a mean difference around 7–13 days (median values: 5–7 days) to the hard dough stage in 2017 (Figure 6). VH and VV backscatter both show a breakpoint as well as a local maximum during the time of the hard dough stage. In contrast to polarimetric parameters, the local maxima are found for both years, whereas the breakpoints are only present in 2018. The local maximum of VV backscatter is located approximately 6–8 days before the hard dough stage (median: 5–9 days) (Figure 13A). In 2018, the fourth breakpoint of VH backscatter has a mean difference of around 3–5 days (median: 1–3.5 days) and the third breakpoint of VV backscatter has a mean difference around 6–8 days (median: 1–3.5 days) to the reported entry date of hard dough. The fourth

breakpoint of NDVI in 2018 results in mean differences of approximately 6.5 days (median: 4.5 days) for Blönsdorf and around 14 days (median: 11 days) for Demmin.

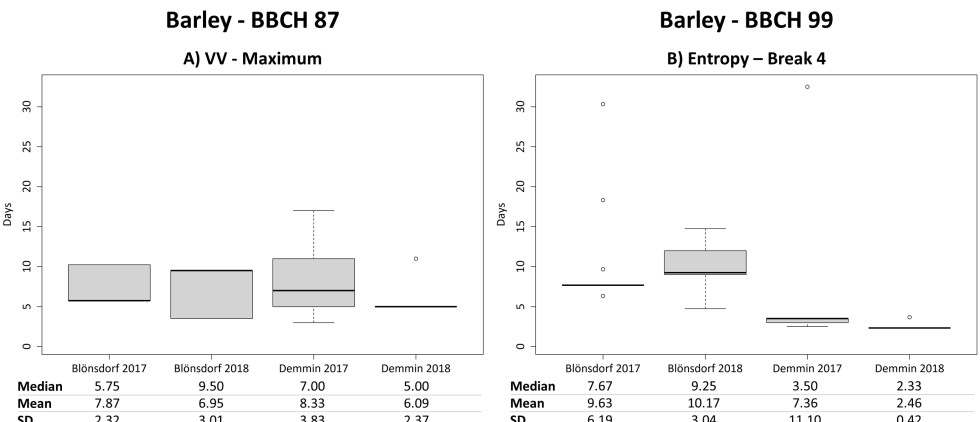

**Figure 13.** Boxplots of temporal differences in days between the local maximum of VV backscatter and the entry date of hard dough (BBCH 87) of barley (**A**) as well as boxplots of temporal differences in days between the fourth breakpoint of entropy and the harvest date (BBCH 99) of barley (**B**).

*4.5. BBCH 99—Harvest*

The harvest date (BBCH 99) is the most drastic change of the field appearance because all plants are usually harvested within one day. The harvest date is quite individually between fields and often depends on availability of machinery and manpower next to weather conditions. In 2017, the harvest of wheat in DEMMIN took only place in August, whereas wheat fields in Blönsdorf are already harvested end of July. In 2018, wheat fields are harvested at the beginning of July (Blönsdorf) or in mid-July (DEMMIN). Barley fields are harvested at the beginning of July 2017 and at the end of June in 2018.

Surprisingly, no parameter reliably indicates the harvest date of wheat. Single parameters show low mean differences between harvest and local extrema or breakpoints, for example a local minimum of the NDVI in Blönsdorf 2017 (Figure 5F). However, this behavior is not observed in further years and test sites.

The harvest of barley fields is detected by the fourth breakpoint of the polarimetric parameters as well as VH backscatter and VH/VV (Figure 6). The mean difference between reported harvest and the fourth breakpoint is around 3–12 days (median: 2–9 days), whereas the difference is generally lower in DEMMIN in both years (Figure 13B). Furthermore, VH backscatter detects a breakpoint around 5–10 days before harvest, whereas VV backscatter reveals a local minimum at approximately 9–14 days (median: 2–10 days). Because of the individual harvest dates of the single fields, mean differences are in some cases influenced by outliers and the median values are better indicators. The NDVI does not reliably indicate the harvest of barley.

## 5. Discussion

*5.1. Evaluation of the Results*

The results of this study confirm and complement results of earlier studies. Schlund and Erasmi [30] and Löw et al. [31] also identified a breakpoint at the beginning of stem elongation. Both studies could furthermore identify a breakpoint during the harvest period for wheat, which was only the case for barley in this study. Similar to our study, Schlund and Erasmi [30] could not detect ripeness with a high degree of certainty. Nasrallah et al. [32] identified the VH/VV ratio as particularly suitable to detect phenological development, whereas in our study, the VH/VV ratio performed equally well as the polarimetric parameters entropy, anisotropy and alpha in most cases.

The evaluation of the results depends on the intended application purpose. In this study, a difference lower than 10 days is considered as a good fit. For farmers, a difference of 10 days between time series feature and entry date of a specific BBCH stage might be

too high for some BBCH stages, whereas the accurate detection of the entry date of the beginning of heading is rather important, knowing the correct date is not that crucial for the beginning of stem elongation or ripening stages. On the other hand, an accuracy of around 10 days might be sufficient for hydrological and climate modeling. Furthermore, a low variability between fields as well as similar results between test sites and years are evaluated as good results. Moreover, the combination of different parameters is helpful. The more parameters are sensitive for a BBCH entry date, the more certain is the result.

The consistency of the results, meaning the majority of fields show a similar difference in days in both test sites and both years, is similarly important than a low difference in days. This is, for example, the case for the beginning of heading (BBCH 51), where the local maxima or minima of polarimetric parameters are located a few days before. In this case, the local extrema might be rather sensitive to a few BBCH stages earlier, for example the development of the flag leave (BBCH 47). Knowing this, the beginning of heading, which is a crucial time step for farmers to apply fungizids, could be predicted.

Additionally, similar results between test sites and years indicate stable results and a successful transferability. Although the analyzed test sites DEMMIN and Blönsdorf do not differ remarkably in their geographic conditions, their comparability is a promising step towards a fully transferable approach of the method. Considering meteorological conditions, the test site Blönsdorf is characterized by higher temperatures and lower precipitation because of its more continental location, which usually leads to a faster plant development of on average around one week compared to DEMMIN. Main reasons for differences between years are varying weather conditions, which was also found by Schlund and Erasmi [30]. However, although the two years 2017 and 2018 are meteorologically extremely different with very dry and hot conditions in 2018 and rather wet conditions in 2017, overall trends are similar for most BBCH stages. The detection of the beginning of stem elongation (BBCH 31) and heading (BBCH 51) show very similar mean differences to corresponding SAR parameters and NDVI. However, some time series features only detect entry dates of BBCH stages for specific years. The beginning of stem elongation (BBCH 31) of barley is detected by more parameters and with a higher accuracy in 2018 because of a higher contribution of soil to the signal. The reason is the slower development because of colder temperatures in early 2018, whereas in 2017 barley plants were already very dense and high at the same time. The medium milk stage (BBCH 75) is only detected for wheat in the wet year 2017. This might be due to the extremely fast ripening in 2018 so that the medium milk stage might not be captured by the sensors. The hard dough stage (BBCH 87) shows different time series features in different years for barley and is only detected in 2018 for wheat. Furthermore, for barley, results are much better in 2018 and captured by more SAR parameters. This suggests that the detection of hard dough can only be performed in dry years, but extended studies with more years are necessary to confirm this assumption.

Differences between wheat and barley are most apparent at the beginning of the vegetation period, when barley is usually more dense and further developed. Therefore, the first instead of the second breakpoint of most time series matches best with the entry date of BBCH 31 (beginning of stem elongation). The reason might be the already higher development of barley at this time with already higher biomass, LAI and plant height. Consequently, the contribution of soil to the signal is lower than that of vegetation for barley, which is detected by all SAR parameters and NDVI. Furthermore, this is only the case for fields of 2018, which are generally less developed compared to 2017. In 2017, barley plants might be already that dense at the time of the beginning of stem elongation, that no explicit change is detected by the parameters. Furthermore, the ripening stage (BBCH 87, hard dough) and the harvest (BBCH 99) is more prominent in barley fields.

There are some reasons why an extreme event like harvest is not clearly detectable by time series of SAR parameters and NDVI, particularly for wheat. One reason is the smoothing of the time series, that eliminates extreme breaks in the curve. Some fields indeed indicate a drastic change of the curve around harvest, which is not visible anymore after smoothing. Another reason is the condition of the field after harvest, which was also

found by Shang et al. [33]. Whereas some fields are completely cleared and ploughed and the signal of the bare soil is influenced by soil moisture, some crop residues in different heights and densities are left on other fields (Figure 3). These different post-harvest characteristics complicate the detection of stable time series features to detect harvest events. Regarding the NDVI, the index indicates photosynthetic activity of plants, which is already very low at the end of the ripening stage and therefore does not necessarily result in drastic differences before and after harvest. However, the fourth breakpoint of multiple SAR parameters is often located between BBCH 87 (hard dough) and BBCH 99 (harvest) (Figures 5 and 6). Similarly to Schlund and Erasmi [30], who also used the last breakpoint to detect harvest, this breakpoint can be an indicator for a near harvest event. For future analyses, the explicit harvest dates of each field can be requested by farmers. In contrast to other phenological entry dates, harvest dates are easy to track and the information is usually available at field level.

### 5.2. Uncertainties and Outlook

Although some mean and median differences in days between reported entry dates of BBCH stages and the corresponding feature of the time series are with less than six days lower than the temporal resolution of Sentinel-1 and Sentinel-2, these numbers are still dependent on the concrete acquisition dates of both sensors. The NDVI performs comparably to SAR parameters, but some disadvantages become apparent. Most evident is the lack of an adequate number of images in 2017. Consequently, it was not possible to smooth the time series and to calculate meaningful breakpoints in this year. However, a global maximum of NDVI is visible around the time of the beginning of heading in both years, but particularly in 2017, the date of this maximum is highly dependent on the acquisition date of Sentinel-2. The irregular availability of Sentinel-2 images in general is a further problem that might prevent the comparability of the findings of this study to other years with a different distribution of suitable images.

Additionally, the smoothing algorithm and particularly the degree of the polynomial regression and the chosen span value, which defines the degree of smoothing, influence the results because the degree of smoothing decisively determines the occurrence and location of local extrema and breakpoints. Particularly, local extrema that do not differ remarkably from their neighboring values might not indicate significant phenological changes and might be not detected by studies using different smoothing algorithms with varying parameters. Additionally, it might be advantageous in future analyses to separate between significant and non-significant local extrema as it was already done by Löw et al. [31] and Meroni et al. [34].

In this study, the detection of phenological stages is analyzed retrospectively for complete vegetation periods. However, farmers might be more interested in having current information about the development of their fields to have the chance to react accordingly, for example, with adapted fertilizing or irrigation strategies. The knowledge about the general temporal behavior of wheat and barley enables to explicitly look for local extrema, exemplary during the time of the expected beginning of heading (BBCH 51). Furthermore, the local maximum of VH backscatter could be used to detect the beginning of stem elongation, which takes place a few days later (Figures 5 and 6). A local maximum of VV and VH backscatter of barley is also detected a few days before hard dough (BBCH 87) and indicates ripening (Figure 6D,E). It is more challenging to calculate breakpoints for incomplete time series. The calculation of breakpoints needs a specific number of observations, therefore the time at which a breakpoint can be clearly identified might be already too late for the farmer. Considering the temporal resolution of six days of Sentinel-1, the identification of a breakpoint can only take place at least two or three observation afterwards, which would be at least around two weeks later. Additionally, the revisit frequency of six days only accounts for Europe and is even lower in other parts of the world.

Another problem of breakpoints are their varying number between fields. In this study, breakpoints are numbered in the order in which they appear. However, as soon as a field has an additional breakpoint at the beginning of the vegetation period or lacks of a breakpoint, the numbering is confused and the transferability between fields is complicated. In this case, the first breakpoint instead of the second one would best detect the beginning of stem elongation, but is not considered in this study. This also explains some outliers, which result from a wrong breakpoint numbering. In future analyses, breakpoints could be explicitly searched in a known time period and unexceptional breakpoints could be ignored.

There are some further uncertainties that might influence the results of this study. A great advantage of the DWD phenology data is their free availability as well as the numerous phenology stations, often offering complete time series of up to 30 years. However, in most cases only one field per phenology station is observed, consequently the analyses of this study are dependent on the report of the entry dates of at most four fields in the best case. These fields are located a few kilometers outside of the test sites and might have slightly differing conditions. Although the selected stations are filtered beforehand by using own phenological observations of single fields of the test sites, and although the mean difference to all phenological stations is considered in this study, some entry dates of the selected phenological stations differ remarkably, which worse the results. Furthermore, the entry dates of a BBCH stage are already reported as soon as half of the field or half of the plants show a specific behavior. In consequence, SAR parameters and NDVI might react a few days later to a specific change, for instance as soon as the whole field underwent this change.

In the future, the detection of within-field heterogeneity of the phenological development would be advantageous to enable a site-specific management for farmers. However, previous studies found the challenge of SAR data to detect differences of biophysical parameters like biomass, LAI or plant height within a field, mainly caused by geometric inaccuracies [16,17]. Furthermore, the next step is to perform the detection of phenological entry dates for larger regions and for further years, exemplary for complete federal states like Mecklenburg-West Pomerania.

## 6. Conclusions

This study confirms that time series features of different SAR parameters as well as NDVI are sensitive to phenological changes and therefore can detect phenological entry dates with different accuracies and transferabilites between years and test sites. The beginning of stem elongation (BBCH 31) of wheat and barley is successfully detected by the first (barley), respectively, the second (wheat) breakpoints and a local maximum of VH backscatter with an accuracy often lower than 9 days. Local minima of anisotropy, VH and VV backscatter as well as local maxima of alpha, entropy, VH/VV and NDVI even better detect the beginning of stem elongation (BBCH 51) for both crop types, whereas VH and VV backscatter show lowest differences of less than 5 days between entry date and minimum for both crop types. The medium milk stage (BBCH 75) cannot be detected with a high degree of certainty and only for 2017. The hard dough stage (BBCH 87) can only be detected in the very dry and hot year 2018 with differences of often less then 5 days by the fourth breakpoint as well as by local extrema of different polarimetric and backscatter parameters. Multiple reasons such as varying field conditions after harvesting as well as the smoothing of the time series complicate the detection of the harvest dates. Harvest cannot be detected for wheat, but with a high accuracy for barley using the fourth breakpoint of polarimetric parameters, VH/VV and VH backscatter as well as a local minimum of VV backscatter.

Differences between test sites and years are mainly caused by varying meteorological conditions, whereas some BBCH stages can only be detected in a rather wet year (BBCH 75), while other BBCH stages can only be detected in very dry years like 2018 (BBCH 87). The variability between years is higher than between test sites, therefore the method is transferable to different test sites as well. For future analyses, the incorporation of further

years as well as of test sites from different regions is suggested to verify the transferability of the method. Backscatter parameters, polarimetric parameters and NDVI complement each other because of their different reactions to phenological changes. However, NDVI time series are restricted to years with a sufficiently high data basis, therefore the transferability between years is questionable in this case because of irregular acquisition dates. The use of dense SAR time series is thus recommendable. In the future, it is aimed to realize a real-time application that can be used by farmers to adapt their management strategies according to the current plant development.

**Author Contributions:** Conceptualization, K.H., S.I., D.S. and C.W.; methodology, K.H.; software, K.H.; validation, K.H.; formal analysis, K.H.; investigation, K.H.; resources, K.H.; data curation, K.H.; writing—original draft preparation, K.H.; writing—review and editing, K.H., S.I., D.S. and C.W.; visualization, K.H.; supervision, S.I., D.S. and C.W.; project administration, D.S.; funding acquisition, D.S. All authors have read and agreed to the published version of the manuscript.

**Funding:** The project is supported by funds of the Federal Ministry of Food and Agriculture (BMEL) based on a decision of the Parliament of the Federal Republic of Germany. The Federal Office for Agriculture and Food (BLE) provides coordinating support for digitalization in agriculture as funding organization, grant numbers 2815710715 (project AgriFusion) and 28DE114A18 (project AgriSens DEMMIN 4.0).

**Data Availability Statement:** Publicly available data sets were analyzed in this study. Sentinel-1 and Sentinel-2 data can be found here: https://scihub.copernicus.eu/ (accessed on 30 October 2021). Phenological data from DWD can be found here: https://opendata.dwd.de/climate_environment/ CDC/observations_germany/phenology/ (accessed on 30 October 2021). Field data and resulting data sets presented in this study are available on request from the corresponding author.

**Conflicts of Interest:** The authors declare no conflict of interest. The funders had no role in the design of the study; in the collection, analyses or interpretation of data; in the writing of the manuscript; or in the decision to publish the results.

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
