# Peer review of "Detecting Phenological Development of Winter Wheat and Winter Barley Using Time Series of Sentinel-1 and Sentinel-2"

_remotesensing, doi:10.3390/rs13245036_

Round 1
Reviewer 1 Report
Dear authors,
Thank you for submitting your interesting study. The paper under evaluation is a sound scientific work, showing the analyses of multiple satellite products to indicate/characterize the phenology of crops over two test sites and for two years. The introduction is overall sound but can be further improved. Objectives are clearly stated but the motivation for the study should be better outlined. Methods and materials are satisfying and follow the state-of-the-art. Results are presented in a good, clear, and structured way. The last section discusses the findings in a balanced way, still some issues need attention. The conclusions are mostly supported by the results, however especially the last sentence is misleading as the approach does not allow for a near-real-time application.
Overall, it is a good, sound, and interesting contribution that needs some few major revisions before publication. Please consider following issues:
- English language and grammar need attention and a thorough proof reading. I am not a native speaker, but especially check the grammar (tempus, commas).
- The introduction needs some enhancement. In the first part more references must be provided on the phenology and its importance. Further, the problem statement is somewhat unclear, so why was this study conducted and why does it provide merit? How does it complement existing studies, e.g. Schlund et Erasmi, Löw et al.? That information is partially already provided but needs sharpening and rephrasing.
- Discussion should also address the applied smoothing and its parameterization. This might indicate also future research needs/directions. Doing so, it might be appropriate to organize the discussion in subsections, which will also enhance the readability.
Besides, please consider the issues listed below.
I look forward to the revised version of the manuscript.
###############################
Further comments:
L1: monitoring the phenological
L3: wording: “of remote sensing parameters” maybe better use “features” or “products” instead of parameters
L4: “two test sites” in northeastern Germany
L5: cancel parameters
L5/6: “smoothed time series of the Sentinel-1 backscatter parameters VH (vertical-horizontal), VV (vertical-vertical) and their ratio VH/VV” maybe just “smoothed time series of the Sentinel-1 VV and VH intensities and their ratio ...”
L6: “parameters” maybe better use “features”
L8: wording: “phenological entry dates” maybe better use “phenological stages/development”
L24: rephrase sentence starting with “Next to crop yield prediction ...”
L20-28: references needed here
L38: rephrase sentence starting with “Furthermore, data”, check tempus
L45: “crop parameters like biomass, plant height and LAI as well.” Include references
L43-48: differences arising from different wavelength should be mentioned here briefly
L60: InSAR coherence? Or PolSAR related coherence like interchannel coherence?
L63: Mercier et al is a classification approach which should be mentioned. Otherwise the reader might think it is also a time series based approach
L64: After this summary, a brief problem statement should be included, e.g. you could state that analysis mostly focused on a single test site or year and that therefore your work is targeting the transferability and repeatability. Compare to information provided in L74-L79
L74 ff: Schlund& Erasmi is a multi year study. Rephrase otherwise misleading
L82: “glacially formed landscape elements” it are rather glacio-fluvial landforms
L85: cancel “already”
L89: hill chain sounds odd
L96: while instead of however
L105: cancel “which is available free of charge”
L108 cancel “only”
L108: is it the mean incidence angle or the center incidence angle of the mid-range? Which subswathes (IW1, IW2, IW3) were processed?
L111: Why only July? The harvest of wheat in Demmin for 2017 happend in early-late August
L115: cancel “as well”
Equation 1: The formular is wrong. Switch numerator and denominator
L126: Abbreviation DWD already introduced?
L129: cancel all
L133: DWD check comment on DWD above
L135: what is about the difference at which an observation was considered “too much”?
L144-L149: Does this cause a problem for the analysis?
L154: “describe”, rather “investigate”?
L167: Here the meaning of Entropy, Alpha and Anisotropy should be explained briefly, also mentioning that the interpretation of these PolSAR parameters differs from the same parameters when deduced from quad-polarimetric data.
L175: It is not clear from the description here, how the ratio was calculated. You calculated the ratio before converting to dB, or you subtracted VV and VH after scaling to dB?
L186: It might also be a higher order polynomial
L189: That means that it is a qualitative assessment of smoothing quality? That should further discussed in the discussion
Figure 4: Each gray line is the time series of a single field/polygon? If possible increase font size
L229: In reference to which stage?
L231: Abbreviation LAI already introduced?
L231 ff: “radar signal is increasingly influenced by vegetation at the expense of surface scattering of the soil” >> “radar signal is presumptively increasingly influenced by volume scattering at the expense of surface scattering” This is an interpretation, you cannot really get this from the dual-pol. data
L244ff: Mean difference between in situ and NDVI or SAR and NDVI?
L251 ff: Mixture between description of results and discussion/ interpretation
L254: less developed
L259: “nearly perfectly” qualitative description for quantitative measurement
L275: onset or start instead of beginning might be better phrasing
L284: General comment on the frequent use of “show”: Usage of a thesaurus recommended: display exhibit,.. Would improve readability significantly
L299: General comment on maximum/minimum: Usage of synonym suggested: extrema, extreme values
L326 ff: What is this marginal difference in value?
L340: citation needed, because the structural volume is still much higher than at the start of season
L352: Maybe include separation of significant and nonsignificat extrema as demonstrated by Meroni et al. (2021), since a marginal change might not be related to actual phenological change
L371-381: Here a timespan is discussed that is partially not covered by the graphs shown in Results. Its a bit confusing and there is no reasoning provided
L401: Why? The explanation is somewhat provided in line 404 ff. So bring those sentences together for a more concise argument.
L411: Please rephrase
L418/419: According to the characteristics provided by the authors the sites do not differ much in their geographic conditions compared to e.g. entire Germany. Hence, a robust assessment of transferability seems unlikely. It is more of a promising start towards a fully transferable approach. This should be reflected by the phrasing
L422/423: A number would provide more context for readers. E.g.: How many days on average is plant development in Blönsdorf ahead of Demmin?
L484/485: 6-days revisit is only applicable to Europe. An additional limitation which is worth mentioning.
L545/546: This last sentence should be canceled. The conducted approach does not allow for a near-real time application that would be needed by the farmers to “adapt their management strategies according to the current plant development”. The approach uses “archive” data and looks on the past development. Compare also to statements in L472 ff.
Author Response
Dear Reviewer,
we would like to thank you very much for reviewing our research article “Detecting Phenological Development of Winter Wheat and Winter Barley Using Time Series of Sentinel-1 and Sentinel-2” authored by Katharina Harfenmeister, Sibylle Itzerott, Cornelia Weltzien and Daniel Spengler.
We carefully read your very valuable and helpful suggestions and questions and revised our manuscript accordingly.
Please find attached the point-by-point responses to your comments as well as the revised manuscript with highlighted changes. Cited line numbers always refer to the revised manuscript.
Sincerely,
Katharina Harfenmeister

Reviewer 2 Report
Generally, it is an interesting work. this study applied multiple types of time series of Sentinel-1 and Sentinel-2 to detect phenological development of winter wheat and winter barley. The experiment is rich in content and data source. Detailed phenological events, such as stem elongation, milk stage, are depicted using SAR backscatter and NDVI data. whereas there are still some issues should be addressed to improve this study.
- the introduction section should be rewritten in a better logical line. Each paragraph in this section should has a theme that will compose the main line of your story. in current case, there are some chaos in introduction section. specially, previous works relevant to this study should not be introduced in the way ‘xxx studied xxx problem’, which is an awkward expression.
- time series of several data is smoothed using LOESS method, please show the performance for it. Are the growth curves in Figure 4 smoothed by this method?
- again, in the discussion section, referenced paper 14-16 is presented as a comparison with this study. Is there any other research in this field?
- there are many abbreviated words for different cases of phenological stage. That makes it hard to thoroughly understand this
- Can case studies for two years field work provide implication for long term study at large spatial range? Different phenological parameters should have diverse applicability in that case. I suggest adding this content in discussion section.

Author Response
Dear Reviewer,
we would like to thank you very much for reviewing our research article “Detecting Phenological Development of Winter Wheat and Winter Barley Using Time Series of Sentinel-1 and Sentinel-2” authored by Katharina Harfenmeister, Sibylle Itzerott, Cornelia Weltzien and Daniel Spengler.
We carefully read your suggestions and questions and revised our manuscript accordingly.
Please find attached the point-by-point responses to your comments as well as the revised manuscript with highlighted changes. Cited line numbers always refer to the revised manuscript.
Sincerely,
Katharina Harfenmeister

Reviewer 3 Report
The monitoring of Crop Phenology is one of the concerns of agricultural remote sensing. Taking winter wheat and winter barley as research objects, the manuscript studied the backscattering and polarization parameters of microwave radar and the relationship between optical vegetation index and crop phenological information. It should be said that this is an interesting work. The results show that backscattering, polarization parameters and NDVI are very sensitive to specific phenological development, and can be better applied to the monitoring of Crop Phenology. Therefore, in general, this paper is innovative and worth publishing in this journal.
Nevertheless, the manuscript still has the following issues:
- Abstract:“This enables the prediction of phenological stages for agricultural monitoring, but differences between test sites and years mainly caused by meteorological differences have to be considered. ”---- Can the results be used to predict crop phenological information? The study shown in the manuscript should be to monitor the phenology of crops.
- In Fig. 3, the difference between the pictures of wheat and barrey harvesting is very obvious. The crop stubble in the wheat harvesting field is very long, while the barley harvesting field seems to have been treated. Is this the conventional harvesting method of the two crops (especially, is this the usual harvesting method for barley)? If not, the characteristic information proposed in the harvesting stage may not be typical.
- The optical image of Sentinel-2 is vulnerable to clouds. Then, during the observation period, it is inevitable that there will be too many clouds and it is difficult to deal with them. How does this study deal with this problem? Please explain in the data preprocessing section.
- At the end of section 4.5, the author said, "the NDVI does not show the harvest of Barclay." what is the reason? It can be seen from Figure 3 that the barley field after harvest is very flat and there is little crop stubble. Why is there no significant change in the NDVI index?
- In the manuscript, most of the results obtained from Monitoring Crop Phenology using different remote sensing information are qualitative analysis results. Can some quantitative analysis results be given (for example, what is the percentage of each growth stage that can be effectively monitored)?
- In the discussion section, the backscattering and polarization parameters and the characteristic signals of NDVI in different phenological stages of crops are discussed in detail. However, the reason why this characteristic signal appears (the reason why this characteristic signal appears) has not been discussed. It is suggested to make appropriate supplement.
- The conclusion is not refined enough, and many contents are the repetition of the results and discussion. It is suggested to further modify and summarize 2-3 conclusions on the existing basis.

Author Response

(The authors gave the same response as above.)

Reviewer 4 Report
Dear Editor
I have the following minor to moderate concerns.
Manuscript language needs to edit for better understanding.
Keywords
Please use upper case for each keyword
- Introduction
Literature review needs to empower by some newest publications.
Indeed, I am not sure about innovative of this work and presented explains.
- Test Sites
Please change title to study sites
That is important to add coordinate systems of two study sites.
Please add reference for presented climate features.
3.1.1. Sentinel-1 Data
Please delete lines 101-107.
- Materials and Methods
Please add a flowchart for the current study.
Fig. 2
I don’t think this is a method.
Figure 4.
I think this is a result. Please move to results part.
4.1. BBCH 31 - Beginning of Stem Elongation
Fig. 4 and Fig. 5 are repeated!
In general, results and discussion are written in a reasonable form.
Author Response

(The authors gave the same response as above.)

Round 2
Reviewer 1 Report
Dear authors,
Thanks for the comprehensive revision of your manuscript. You addressed all the points I raised in my last report in a very good and reasonable manner. In my opinion, the quality of the manuscript has increased considerably, and your contribution is now ready for publication.
I have very few notes on the revised version (Version 2), which I suppose can be addressed during the final round and/or production:
(1) Line 401-403: “days before harvest” please briefly indicate here, that for DEMMIN that information is taken from 2018 only, as “... the harvest of wheat in DEMMIN took only place in August, ...” (Line 391/392)
(2) Line 356-358; “The LAI decreases and the soil has again an increasing influence on the radar signal or the reflection, whereas the plant volume is still higher than at the beginning of the season.” I suggest making this statement less strong, e.g. “... and the soil has, presumptively/likely again ...”. Finally, you cannot infer this from the S-1 record.
(3) Last sentence Line 589-591: This statement is still somehow contradictive to your statement in Line 522ff. If several acquisitions are needed to detect a breakpoint, the method cannot be a near-real time application. My suggestion is to cancel “... of the method ...” in this sentence.
Congratulations and thanks for your interesting work.
Stay well and kind regards
Author Response
Dear Reviewer,
thanks a lot for your comments! We changed and/or answered to the three points you raised in the second revision round. Please find attached the new version of the manuscript with highlighted changes.
(1) Lines 402-404 (previous lines 401-403) refer to the harvest of barley, which took place in June and July in both years (new line 397), therefore the information is also taken from 2017. We have not changed anything here and hope that we have not misunderstood your point.
(2) We added “presumably” in line 359 to extenuate the statement.
(3) We removed “of the method” in the last sentence. Thanks for pointing this out!
According to the suggestions of another reviewer, we furthermore added the following sentence in the introduction in lines 39-41: “Additionally, time-lapse cameras installed close to the surface (PhenoCams) are used to track phenology at very high temporal resolutions (e.g. at a daily rate) but are mostly limited to the field scale [11,12].”
Furthermore, we tried to arrange the Figures and Tables a bit more advantageously, but the final placement will be probably made during the finalization of the manuscript by the editors.
Thanks again for your very valuable evaluation of our manuscript!
Best regards,
Katharina Harfenmeister

Reviewer 2 Report
The authors have responded to my comments point-to-point and explained my doubts. In current status, the logic of this study has made sense and can be understood smoothly. Formats for figure and tables should be adjusted, such as Figure 6, Table 3. I suggest adding some relevant reference papers on near surface phenological monitoring, such as phenocam studies over the world.
This revision satisfied my requirements. It can be published on Remote Sensing after minor revision.
Author Response
Dear Reviewer,
thanks again for reviewing our manuscript!
According to your suggestions, we added two references of near surface phenological monitoring in the introduction in lines 39-41: “Additionally, time-lapse cameras installed close to the surface (PhenoCams) are used to track phenology at very high temporal resolutions (e.g. at a daily rate) but are mostly limited to the field scale [11,12].”
We assume that the final placement and size of the Figures and Tables is made during the finalization of the manuscript by the editors, this was at least the case with our earlier publications. However, we tried to arrange them already a bit more advantageously.
According to the suggestions of another reviewer, we furthermore made two small changes in the following lines:
Line 359: We added “presumably” to extenuate the statement.
Line 592: We removed “of the method” in the last sentence of the conclusion.
Please find attached the new version of the manuscript with highlighted changes.
Best regards,
Katharina Harfenmeister

Reviewer 3 Report
The author has modified the manuscript as required, and has explained in detail what has not been modified. Recommended for publication.
Author Response
Dear Reviewer,
thanks a lot for your assessment. According to two other reviewers, the following changes were made to the manuscript during the second revision round:
- Line 359: We added “presumably” to extenuate the statement.
- Line 592: We removed “of the method” in the last sentence of the conclusion.
- We added the following sentence in the introduction in lines 39-41: “Additionally, time-lapse cameras installed close to the surface (PhenoCams) are used to track phenology at very high temporal resolutions (e.g. at a daily rate) but are mostly limited to the field scale [11,12].”
- We tried to arrange the Figures and Tables a bit more advantageously.
Please find attached the revised manuscript with highlighted changes.
Best regards,
Katharina Harfenmeister
